# Learning Latent Space Energy-Based Prior Model

**Bo Pang**[*1]   **Tian Han**[*2]   **Erik Nijkamp**[*1]   **Song-Chun Zhu**[1]   **Ying Nian Wu**[1]
[1]University of California, Los Angeles    [2]Stevens Institute of Technology
{bopang, enijkamp}@ucla.edu    than6@stevens.edu    {sczhu, ywu}@stat.ucla.edu

## Abstract

We propose to learn energy-based model (EBM) in the latent space of a generator model, so that the EBM serves as a prior model that stands on the top-down network of the generator model. Both the latent space EBM and the top-down network can be learned jointly by maximum likelihood, which involves short-run MCMC sampling from both the prior and posterior distributions of the latent vector. Due to the low dimensionality of the latent space and the expressiveness of the top-down network, a simple EBM in latent space can capture regularities in the data effectively, and MCMC sampling in latent space is efficient and mixes well. We show that the learned model exhibits strong performances in terms of image and text generation and anomaly detection. *The one-page code can be found in supplementary materials.*

## 1   Introduction

In recent years, deep generative models have achieved impressive successes in image and text generation. A particularly simple and powerful model is the generator model [35, 21], which assumes that the observed example is generated by a low-dimensional latent vector via a top-down network, and the latent vector follows a non-informative prior distribution, such as uniform or isotropic Gaussian distribution. While we can learn an expressive top-down network to map the prior distribution to the data distribution, we can also learn an informative prior model in the latent space to further improve the expressive power of the whole model. This follows the philosophy of empirical Bayes where the prior model is learned from the observed data. Specifically, we assume the latent vector follows an energy-based model (EBM). We call this model the latent space energy-based prior model.

Both the latent space EBM and the top-down network can be learned jointly by maximum likelihood estimate (MLE). Each learning iteration involves Markov chain Monte Carlo (MCMC) sampling of the latent vector from both the prior and posterior distributions. Parameters of the prior model can then be updated based on the statistical difference between samples from the two distributions. Parameters of the top-down network can be updated based on the samples from the posterior distribution as well as the observed data.

Due to the low-dimensionality of the latent space, the energy function can be parametrized by a small multi-layer perceptron, yet the energy function can capture regularities in the data effectively because the EBM stands on an expressive top-down network. Moreover, MCMC in the latent space for both prior and posterior sampling is efficient and mixes well. Specifically, we employ short-run MCMC [51, 50, 52, 25] which runs a fixed number of steps from a fixed initial distribution. We formulate the resulting learning algorithm as a perturbation of MLE learning in terms of both objective function and estimating equation, so that the learning algorithm has a solid theoretical foundation. Within our theoretical framework, the short-run MCMC for posterior and prior sampling can also be amortized by jointly learned inference and synthesis networks. However, in this initial paper, we prefer keeping our model and learning method pure and self-contained, without mixing in learning

---

[*]Equal contribution.

tricks from variational auto-encoder (VAE) [35, 58] and generative adversarial networks (GAN) [21, 56]. Thus we shall rely on short-run MCMC for simplicity. *The one-page code can be found in supplementary materials.* See our follow-up development on amortized inference in the context of semi-supervised learning [55].

We test the proposed modeling, learning and computing method on tasks such as image synthesis, text generation, as well as anomaly detection. We show that our method is competitive with prior art. See also our follow-up work on molecule generation [54].

**Contributions**. (1) We propose a latent space energy-based prior model that stands on the top-down network of the generator model. (2) We develop the maximum likelihood learning algorithm that learns the EBM prior and the top-down network jointly based on MCMC sampling of the latent vector from the prior and posterior distributions. (3) We further develop an efficient modification of MLE learning based on short-run MCMC sampling. (4) We provide theoretical foundation for learning based on short-run MCMC. The theoretical formulation can also be used to amortize short-run MCMC by extra inference and synthesis networks. (5) We provide strong empirical results to illustrate the proposed method.

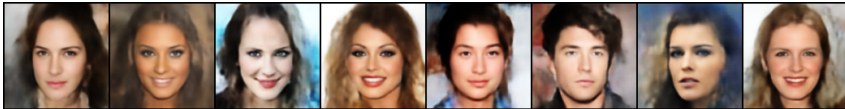

Figure 1: Generated images for CelebA ($128 \times 128 \times 3$).

## 2   Model and learning

### 2.1   Model

Let $x$ be an observed example such as an image or a piece of text, and let $z \in \mathbb{R}^d$ be the latent variables. The joint distribution of $(x, z)$ is

$$p_\theta(x, z) = p_\alpha(z)p_\beta(x|z), \tag{1}$$

where $p_\alpha(z)$ is the prior model with parameters $\alpha$, $p_\beta(x|z)$ is the top-down generation model with parameters $\beta$, and $\theta = (\alpha, \beta)$.

The prior model $p_\alpha(z)$ is formulated as an energy-based model,

$$p_\alpha(z) = \frac{1}{Z(\alpha)} \exp(f_\alpha(z))p_0(z). \tag{2}$$

where $p_0(z)$ is a known reference distribution, assumed to be isotropic Gaussian in this paper. $f_\alpha(z)$ is the negative energy and is parameterized by a small multi-layer perceptron with parameters $\alpha$. $Z(\alpha) = \int \exp(f_\alpha(z))p_0(z)dz = \mathrm{E}_{p_0}[\exp(f_\alpha(z))]$ is the normalizing constant or partition function.

The prior model (2) can be interpreted as an energy-based correction or exponential tilting of the original prior distribution $p_0$, which is the prior distribution in the generator model in VAE.

The generation model is the same as the top-down network in VAE. For image modeling, assuming $x \in \mathbb{R}^D$,

$$x = g_\beta(z) + \epsilon, \tag{3}$$

where $\epsilon \sim \mathrm{N}(0, \sigma^2 I_D)$, so that $p_\beta(x|z) \sim \mathrm{N}(g_\beta(z), \sigma^2 I_D)$. As in VAE, $\sigma^2$ takes an assumed value. For text modeling, let $x = (x^{(t)}, t = 1, ..., T)$ where each $x^{(t)}$ is a token. Following previous text VAE model [6], we define $p_\beta(x|z)$ as a conditional autoregressive model,

$$p_\beta(x|z) = \prod_{t=1}^{T} p_\beta(x^{(t)}|x^{(1)}, ..., x^{(t-1)}, z) \tag{4}$$

which is parameterized by a recurrent network with parameters $\beta$.

In the original generator model, the top-down network $g_\beta$ maps the unimodal prior distribution $p_0$ to be close to the usually highly multi-modal data distribution. The prior model in (2) refines $p_0$ so that

$g_\beta$ maps the prior model $p_\alpha$ to be closer to the data distribution. The prior model $p_\alpha$ does not need to be highly multi-modal because of the expressiveness of $g_\beta$.

The marginal distribution is $p_\theta(x) = \int p_\theta(x, z)dz = \int p_\alpha(z)p_\beta(x|z)dz$. The posterior distribution is $p_\theta(z|x) = p_\theta(x, z)/p_\theta(x) = p_\alpha(z)p_\beta(x|z)/p_\theta(x)$.

In the above model, we exponentially tilt $p_0(z)$. We can also exponentially tilt $p_0(x, z) = p_0(z)p_\beta(x|z)$ to $p_\theta(x, z) = \frac{1}{Z(\theta)}\exp(f_\alpha(x, z))p_0(x, z)$. Equivalently, we may also exponentially tilt $p_0(z, \epsilon) = p_0(z)p(\epsilon)$, as the mapping from $(z, \epsilon)$ to $(z, x)$ is a change of variable. This leads to an EBM in both the latent space and data space, which makes learning and sampling more complex. Therefore, we choose to only tilt $p_0(z)$ and leave $p_\beta(x|z)$ as a directed top-down generation model.

## 2.2 Maximum likelihood

Suppose we observe training examples $(x_i, i = 1, ..., n)$. The log-likelihood function is

$$L(\theta) = \sum_{i=1}^{n} \log p_\theta(x_i). \tag{5}$$

The learning gradient can be calculated according to

$$\nabla_\theta \log p_\theta(x) = \mathrm{E}_{p_\theta(z|x)}\left[\nabla_\theta \log p_\theta(x, z)\right] = \mathrm{E}_{p_\theta(z|x)}\left[\nabla_\theta(\log p_\alpha(z) + \log p_\beta(x|z))\right]. \tag{6}$$

See Theoretical derivations in the Supplementary for a detailed derivation.

For the prior model, $\nabla_\alpha \log p_\alpha(z) = \nabla_\alpha f_\alpha(z) - \mathrm{E}_{p_\alpha(z)}[\nabla_\alpha f_\alpha(z)]$. Thus the learning gradient for an example $x$ is

$$\delta_\alpha(x) = \nabla_\alpha \log p_\theta(x) = \mathrm{E}_{p_\theta(z|x)}[\nabla_\alpha f_\alpha(z)] - \mathrm{E}_{p_\alpha(z)}[\nabla_\alpha f_\alpha(z)]. \tag{7}$$

The above equation has an empirical Bayes nature. $p_\theta(z|x)$ is based on the empirical observation $x$, while $p_\alpha$ is the prior model. $\alpha$ is updated based on the difference between $z$ inferred from empirical observation $x$, and $z$ sampled from the current prior.

For the generation model,

$$\delta_\beta(x) = \nabla_\beta \log p_\theta(x) = \mathrm{E}_{p_\theta(z|x)}[\nabla_\beta \log p_\beta(x|z)], \tag{8}$$

where $\log p_\beta(x|z) = -\|x - g_\beta(z)\|^2/(2\sigma^2) + \text{const}$ or $\sum_{t=1}^{T} \log p_\beta(x^{(t)}|x^{(1)}, ..., x^{(t-1)}, z)$ for image and text modeling respectively.

Expectations in (7) and (8) require MCMC sampling of the prior model $p_\alpha(z)$ and the posterior distribution $p_\theta(z|x)$. We can use Langevin dynamics [47, 82]. For a target distribution $\pi(z)$, the dynamics iterates $z_{k+1} = z_k + s\nabla_z \log \pi(z_k) + \sqrt{2s}\epsilon_k$, where $k$ indexes the time step of the Langevin dynamics, $s$ is a small step size, and $\epsilon_k \sim \mathrm{N}(0, I_d)$ is the Gaussian white noise. $\pi(z)$ can be either $p_\alpha(z)$ or $p_\theta(z|x)$. In either case, $\nabla_z \log \pi(z)$ can be efficiently computed by back-propagation.

## 2.3 Short-run MCMC

Convergence of Langevin dynamics to the target distribution requires infinite steps with infinitesimal step size, which is impractical. We thus propose to use short-run MCMC [51, 50, 52] for approximate sampling.

The short-run Langevin dynamics is always initialized from the fixed initial distribution $p_0$, and only runs a fixed number of $K$ steps, e.g., $K = 20$,

$$z_0 \sim p_0(z), \ z_{k+1} = z_k + s\nabla_z \log \pi(z_k) + \sqrt{2s}\epsilon_k, \ k = 1, ..., K. \tag{9}$$

Denote the distribution of $z_K$ to be $\tilde{\pi}(z)$. Because of fixed $p_0(z)$ and fixed $K$ and $s$, the distribution $\tilde{\pi}$ is well defined. In this paper, we put ˜ sign on top of the symbols to denote distributions or quantities produced by short-run MCMC, and for simplicity, we omit the dependence on $K$ and $s$ in notation. As shown in [9], the Kullback-Leibler divergence $D_{KL}(\tilde{\pi}\|\pi)$ decreases to zero monotonically as $K \to \infty$.

Specifically, denote the distribution of $z_K$ to be $\tilde{p}_\alpha(z)$ if the target $\pi(z) = p_\alpha(z)$, and denote the distribution of $z_K$ to be $\tilde{p}_\theta(z|x)$ if $\pi(z) = p_\theta(z|x)$. We can then replace $p_\alpha(z)$ by $\tilde{p}_\alpha(z)$ and replace

$p_\theta(z|x)$ by $\tilde{p}_\theta(z|x)$ in equations (7) and (8), so that the learning gradients in equations (7) and (8) are modified to

$$\tilde{\delta}_\alpha(x) = \mathrm{E}_{\tilde{p}_\theta(z|x)}[\nabla_\alpha f_\alpha(z)] - \mathrm{E}_{\tilde{p}_\alpha(z)}[\nabla_\alpha f_\alpha(z)], \tag{10}$$

$$\tilde{\delta}_\beta(x) = \mathrm{E}_{\tilde{p}_\theta(z|x)}[\nabla_\beta \log p_\beta(x|z)]. \tag{11}$$

We then update $\alpha$ and $\beta$ based on (10) and (11), where the expectations can be approximated by Monte Carlo samples. See our follow-up work [55] on persistent chains for prior sampling.

## 2.4 Algorithm

The learning and sampling algorithm is described in Algorithm 1.

---

**Algorithm 1:** Learning latent space EBM prior via short-run MCMC.

---

**input** : Learning iterations $T$, learning rate for prior model $\eta_0$, learning rate for generation model $\eta_1$, initial parameters $\theta_0 = (\alpha_0, \beta_0)$, observed examples $\{x_i\}_{i=1}^n$, batch size $m$, number of prior and posterior sampling steps $\{K_0, K_1\}$, and prior and posterior sampling step sizes $\{s_0, s_1\}$.

**output :** $\theta_T = (\alpha_T, \beta_T)$.

**for** $t = 0 : T - 1$ **do**

    1. **Mini-batch**: Sample observed examples $\{x_i\}_{i=1}^m$.

    2. **Prior sampling**: For each $x_i$, sample $z_i^- \sim \tilde{p}_{\alpha_t}(z)$ using equation (9), where the target distribution $\pi(z) = p_{\alpha_t}(z)$, and $s = s_0$, $K = K_0$.

    3. **Posterior sampling**: For each $x_i$, sample $z_i^+ \sim \tilde{p}_{\theta_t}(z|x_i)$ using equation (9), where the target distribution $\pi(z) = p_{\theta_t}(z|x_i)$, and $s = s_1$, $K = K_1$.

    4. **Learning prior model**: $\alpha_{t+1} = \alpha_t + \eta_0 \frac{1}{m} \sum_{i=1}^m [\nabla_\alpha f_{\alpha_t}(z_i^+) - \nabla_\alpha f_{\alpha_t}(z_i^-)]$.

    5. **Learning generation model**: $\beta_{t+1} = \beta_t + \eta_1 \frac{1}{m} \sum_{i=1}^m \nabla_\beta \log p_{\beta_t}(x_i|z_i^+)$.

---

The posterior sampling and prior sampling correspond to the positive phase and negative phase of latent EBM [1].

## 2.5 Theoretical understanding

The learning algorithm based on short-run MCMC sampling in Algorithm 1 is a modification or perturbation of maximum likelihood learning, where we replace $p_\alpha(z)$ and $p_\theta(z|x)$ by $\tilde{p}_\alpha(z)$ and $\tilde{p}_\theta(z|x)$ respectively. For theoretical underpinning, we should understand this perturbation in terms of objective function and estimating equation.

In terms of objective function, define the Kullback-Leibler divergence $D_{KL}(p(x)\|q(x)) = \mathrm{E}_p[\log(p(x)/q(x)]$. At iteration $t$, with fixed $\theta_t = (\alpha_t, \beta_t)$, consider the following computationally tractable perturbation of the log-likelihood function of $\theta$ for an observation $x$,

$$\tilde{l}_\theta(x) = \log p_\theta(x) - D_{KL}(\tilde{p}_{\theta_t}(z|x)\|p_\theta(z|x)) + D_{KL}(\tilde{p}_{\alpha_t}(z)\|p_\alpha(z)). \tag{12}$$

The above is a function of $\theta$, while $\theta_t$ is fixed. Then

$$\tilde{\delta}_\alpha(x) = \nabla_\alpha \tilde{l}_\theta(x), \ \tilde{\delta}_\beta(x) = \nabla_\beta \tilde{l}_\theta(x), \tag{13}$$

where the derivative is taken at $\theta_t$. See Theoretical derivations in the Supplementary for details. Thus the updating rule of Algorithm 1 follows the stochastic gradient (i.e., Monte Carlo approximation of the gradient) of a perturbation of the log-likelihood. Because $\theta_t$ is fixed, we can drop the entropies of $\tilde{p}_{\theta_t}(z|x)$ and $\tilde{p}_{\alpha_t}(z)$ in the above Kullback-Leibler divergences, hence the updating rule follows the stochastic gradient of

$$Q(\theta) = L(\theta) + \sum_{i=1}^n \left[ \mathrm{E}_{\tilde{p}_{\theta_t}(z_i|x_i)}[\log p_\theta(z_i|x_i)] - \mathrm{E}_{\tilde{p}_{\alpha_t}(z)}[\log p_\alpha(z)] \right], \tag{14}$$

where $L(\theta)$ is the total log-likelihood defined in equation (5), and the gradient is taken at $\theta_t$.

In equation (12), the first $D_{KL}$ term is related to the EM algorithm [14]. It leads to the more tractable complete-data log-likelihood. The second $D_{KL}$ term is related to contrastive divergence [64], except that the short-run MCMC for $\tilde{p}_{\alpha_t}(z)$ is initialized from $p_0(z)$. It serves to cancel the intractable $\log Z(\alpha)$ term.

In terms of estimating equation, the stochastic gradient descent in Algorithm 1 is a Robbins-Monro stochastic approximation algorithm [59] that solves the following estimating equation:

$$\frac{1}{n}\sum_{i=1}^{n}\tilde{\delta}_{\alpha}(x_i) = \frac{1}{n}\sum_{i=1}^{n}\mathrm{E}_{\tilde{p}_{\theta}(z_i|x_i)}[\nabla_{\alpha}f_{\alpha}(z_i)] - \mathrm{E}_{\tilde{p}_{\alpha}(z)}[\nabla_{\alpha}f_{\alpha}(z)] = 0, \qquad (15)$$

$$\frac{1}{n}\sum_{i=1}^{n}\tilde{\delta}_{\beta}(x_i) = \frac{1}{n}\sum_{i=1}^{n}\mathrm{E}_{\tilde{p}_{\theta}(z_i|x_i)}[\nabla_{\beta}\log p_{\beta}(x_i|z_i)] = 0. \qquad (16)$$

The solution to the above estimating equation defines an estimator of the parameters. Algorithm 1 converges to this estimator under the usual regularity conditions of Robbins-Monro [59]. If we replace $\tilde{p}_{\alpha}(z)$ by $p_{\alpha}(z)$, and $\tilde{p}_{\theta}(z|x)$ by $p_{\theta}(z|x)$, then the above estimating equation is the maximum likelihood estimating equation.

## 2.6 Amortized inference and synthesis

We can amortize the short-run MCMC sampling of the prior and posterior distributions of the latent vector by jointly learning an extra inference network $q_{\phi}(z|x)$ and an extra synthesis network $q_{\psi}(z)$, together with the original model. Let us re-define $\tilde{l}_{\theta}(x)$ in (12) by

$$\tilde{l}_{\theta,\phi,\psi}(x) = \log p_{\theta}(x) - D_{KL}(q_{\phi}(z|x)\|p_{\theta}(z|x)) + D_{KL}(q_{\psi}(z)\|p_{\alpha}(z)), \qquad (17)$$

where we replace $\tilde{p}_{\theta_t}(z|x)$ in (12) by $q_{\phi}(z|x)$ and replace $\tilde{p}_{\alpha_t}(z)$ in (12) by $q_{\psi}(z)$. See [27, 28] for related formulations. Define $\tilde{L}(\theta, \phi, \psi) = \frac{1}{n}\sum_{i=1}^{n}\tilde{l}_{\theta,\phi,\psi}(x)$, we can jointly learn $(\theta, \phi, \psi)$ by $\max_{\theta,\phi}\min_{\psi}\tilde{L}(\theta, \phi, \psi)$. The objective function $\tilde{L}(\theta, \phi, \psi)$ is a perturbation of the log-likelihood $L(\theta)$ in (5), where $-D_{KL}(q_{\phi}(z|x)\|p_{\theta}(z|x))$ leads to variational learning, and the learning of the inference network $q_{\phi}(z|x)$ follows VAE, except that we include the EBM prior $\log p_{\alpha}(z)$ in training $q_{\phi}(z|x)$ ($\log Z(\alpha)$ can be discarded as a constant relative to $\phi$). The synthesis network $q_{\psi}(z)$ can be taken to be a flow-based model [15, 57]. $D_{KL}(q_{\psi}(z)\|p_{\alpha}(z))$ leads to adversarial training of $q_{\psi}(z)$ and $p_{\alpha}(z)$. $q_{\psi}(z)$ is trained as a variational approximation to $p_{\alpha}(z)$ (again $\log Z(\alpha)$ can be discarded as a constant relative to $\psi$), while $p_{\alpha}(z)$ is updated based on statistical difference between samples from the approximate posterior $q_{\phi}(z|x)$ and samples from the approximate prior $q_{\psi}(z)$, i.e., $p_{\alpha}(z)$ is a critic of $q_{\psi}(z)$. See supplementary materials for a formulation based on three $D_{KL}$ terms.

In this initial paper, we prefer keeping our model and learning method clean and simple, without involving extra networks for learned computations, and without mixing in learning tricks from VAE and GAN. See our follow-up work on joint training of amortized inference network [55]. See also [74] for a temporal difference MCMC teaching scheme for amortizing MCMC.

## 3 Experiments

We present a set of experiments which highlight the effectiveness of our proposed model with (1) excellent synthesis for both visual and textual data outperforming state-of-the-art baselines, (2) high expressiveness of the learned prior model for both data modalities, and (3) strong performance in anomaly detection. For image data, we include SVHN [48], CelebA [42], and CIFAR-10 [36]. For text data, we include PTB [46], Yahoo [78], and SNLI [5]. We refer to the Supplementary for details. Code to reproduce the reported results is available [2]. Recently we extend our work to construct a symbol-vector coupling model for semi-supervised learning and learn it with amortized inference for posterior inference and persistent chains for prior sampling [55], which demonstrates promising results in multiple data domains. In another followup [54], we find the latent space EBM can learn to capture complex chemical laws automatically and implicitly, enabling valid, novel, and diverse molecule generations. Besides the results detailed below, our extended experiments also corroborate our modeling strategy of building a latent space EBM for powerful generative modeling, meaningful representation learning, and stable training.

### 3.1 Image modeling

We evaluate the quality of the generated and reconstructed images. If the model is well-learned, the latent space EBM $\pi_{\alpha}(z)$ will fit the generator posterior $p_{\theta}(z|x)$ which in turn renders realistic

generated samples as well as faithful reconstructions. We compare our model with VAE [35] and SRI [51] which assume a fixed Gaussian prior distribution for the latent vector and two recent strong VAE variants, 2sVAE [10] and RAE [20], whose prior distributions are learned with posterior samples in a second stage. We also compare with multi-layer generator (i.e., 5 layers of latent vectors) model [51] which admits a powerful learned prior on the bottom layer of latent vector. We follow the protocol as in [51].

**Generation**. The generator network $p_\theta$ in our framework is well-learned to generate samples that are realistic and share visual similarities as the training data. The qualitative results are shown in Figure 2. We further evaluate our model quantitatively by using Fréchet Inception Distance (FID) [44] in Table 1. It can be seen that our model achieves superior generation performance compared to listed baseline models.

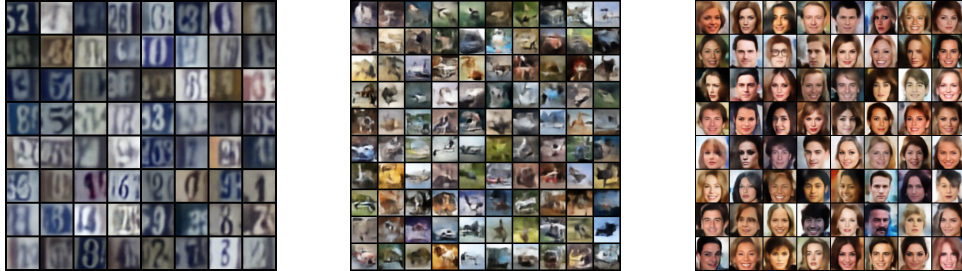

Figure 2: Generated samples for SVHN ($32 \times 32 \times 3$), CIFAR-10 ($32 \times 32 \times 3$), and CelebA ($64 \times 64 \times 3$).

**Reconstruction**. We evaluate the accuracy of the posterior inference by testing image reconstruction. The well-formed posterior Langevin should not only help to learn the latent space EBM model but also match the true posterior $p_\theta(z|x)$ of the generator model. We quantitatively compare reconstructions of test images with the above baseline models on mean square error (MSE). From Table 1, our proposed model could achieve not only high generation quality but also accurate reconstructions.

| Models | | VAE | 2sVAE | RAE | SRI | SRI (L=5) | Ours |
|---|---|---|---|---|---|---|---|
| SVHN | MSE | 0.019 | 0.019 | 0.014 | 0.018 | 0.011 | **0.008** |
| | FID | 46.78 | 42.81 | 40.02 | 44.86 | 35.23 | **29.44** |
| CIFAR-10 | MSE | 0.057 | 0.056 | 0.027 | - | - | **0.020** |
| | FID | 106.37 | 72.90 | 74.16 | - | - | **70.15** |
| CelebA | MSE | 0.021 | 0.021 | 0.018 | 0.020 | 0.015 | **0.013** |
| | FID | 65.75 | 44.40 | 40.95 | 61.03 | 47.95 | **37.87** |

Table 1: MSE of testing reconstructions and FID of generated samples for SVHN ($32 \times 32 \times 3$), CIFAR-10 ($32 \times 32 \times 3$), and CelebA ($64 \times 64 \times 3$) datasets.

## 3.2 Text modeling

We compare our model to related baselines, SA-VAE [34], FB-VAE [41], and ARAE [80]. SA-VAE optimized posterior samples with gradient descent guided by EBLO, resembling the short run dynamics in our model. FB-VAE is the SOTA VAE for text modeling. While SA-VAE and FB-VAE assume a fixed Gaussian prior, ARAE adversarially learns a latent sample generator as an implicit prior distribution. To evaluate the quality of the generated samples, we follow [80, 8] and recruit Forward Perplexity (FPPL) and Reverse Perplexity (RPPL). FPPL is the perplexity of the generated samples evaluated under a language model trained with real data and measures the fluency of the synthesized sentences. RPPL is the perplexity of real data computed under a language model trained with the model-generated samples. Prior work employs it to measure the distributional coverage of a learned model, $p_\theta(x)$ in our case, since a model with a mode-collapsing issue results in a high RPPL. FPPL and RPPL are displayed in Table 2. Our model outperforms all the baselines on the two metrics, demonstrating the high fluency and diversity of the samples from our model. We also evaluate the reconstruction of our model against the baselines using negative log-likelihood (NLL). Our model has a similar performance as that of FB-VAE and ARAE, while they all outperform SA-VAE.

| Models | SNLI | | | PTB | | | Yahoo | | |
|---|---|---|---|---|---|---|---|---|---|
| | FPPL | RPPL | NLL | FPPL | RPPL | NLL | FPPL | RPPL | NLL |
| Real Data | 23.53 | - | - | 100.36 | - | - | 60.04 | - | - |
| SA-VAE | 39.03 | 46.43 | 33.56 | 147.92 | 210.02 | 101.28 | 128.19 | 148.57 | 326.70 |
| FB-VAE | 39.19 | 43.47 | 28.82 | 145.32 | 204.11 | 92.89 | 123.22 | 141.14 | 319.96 |
| ARAE | 44.30 | 82.20 | 28.14 | 165.23 | 232.93 | 91.31 | 158.37 | 216.77 | 320.09 |
| Ours | **27.81** | **31.96** | 28.90 | **107.45** | **181.54** | 91.35 | **80.91** | **118.08** | 321.18 |

Table 2: FPPL, RPPL, and NLL for our model and baselines on SNLI, PTB, and Yahoo datasets.

### 3.3 Analysis of latent space

We examine the exponential tilting of the reference prior $p_0(z)$ through Langevin samples initialized from $p_0(z)$ with target distribution $p_\alpha(z)$. As the reference distribution $p_0(z)$ is in the form of an isotropic Gaussian, we expect the energy-based correction $f_\alpha$ to tilt $p_0$ into an irregular shape. In particular, learning equation 10 may form shallow local modes for $p_\alpha(z)$. Therefore, the trajectory of a Markov chain initialized from the reference distribution $p_0(z)$ with well-learned target $p_\alpha(z)$ should depict the transition towards synthesized examples of high quality while the energy fluctuates around some constant. Figure 3 and Table 3 depict such transitions for image and textual data, respectively, which are both based on models trained with $K_0 = 40$ steps. For image data the quality of synthesis improve significantly with increasing number of steps. For textual data, there is an enhancement in semantics and syntax along the chain, which is especially clear from step 0 to 40 (see Table 3).

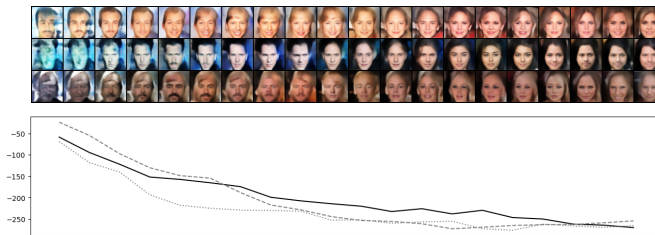

Figure 3: Transition of Markov chains initialized from $p_0(z)$ towards $\tilde{p}_\alpha(z)$ for $K_0' = 100$ steps. *Top:* Trajectory in the CelebA data-space. *Bottom:* Energy profile over time.

| |
|---|
| judge in \<unk\> was not |
| west virginia bank \<unk\> which has been under N law took effect of october N |
| mr. peterson N years old could return to work with his clients to pay |
| iras must be |
| anticipating bonds tied to the imperial company 's revenue of $ N million today |
| many of these N funds in the industrial average rose to N N from N N N |
| fund obtaining the the |
| ford 's latest move is expected to reach an agreement in principle for the sale of its loan operations |
| wall street has been shocked over by the merger of new york co. a world-wide financial board of the companies said it wo |
| n't seek strategic alternatives to the brokerage industry 's directors |

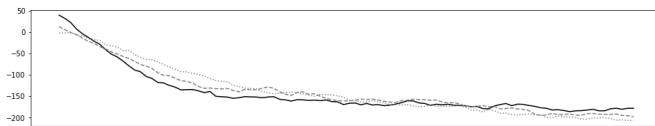

Table 3: Transition of a Markov chain initialized from $p_0(z)$ towards $\tilde{p}_\alpha(z)$. *Top:* Trajectory in the PTB data-space. Each panel contains a sample for $K_0' \in \{0, 40, 100\}$. *Bottom:* Energy profile.

While our learning algorithm recruits short run MCMC with $K_0$ steps to sample from target distribution $p_\alpha(z)$, a well-learned $p_\alpha(z)$ should allow for Markov chains with realistic synthesis for $K_0' \gg K_0$ steps. We demonstrate such long-run Markov chain with $K_0 = 40$ and $K_0' = 2500$ in Figure 4. The long-run chain samples in the data space are reasonable and do not exhibit the oversaturing issue of the long-run chain samples of recent EBM in the data space (see oversaturing examples in Figure 3 in [50]).

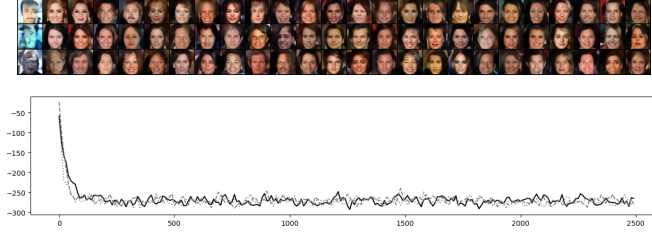

Figure 4: Transition of Markov chains initialized from $p_0(z)$ towards $\tilde{p}_\alpha(z)$ for $K_0' = 2500$ steps. *Top:* Trajectory in the CelebA data-space for every 100 steps. *Bottom:* Energy profile over time.

## 3.4 Anomaly detection

We evaluate our model on anomaly detection. If the generator and EBM are well learned, then the posterior $p_\theta(z|x)$ would form a discriminative latent space that has separated probability densities for normal and anomalous data. Samples from such a latent space can then be used to detect anomalies. We take samples from the posterior of the learned model, and use the unnormalized log-posterior $\log p_\theta(x, z)$ as our decision function.

Following the protocol as in [37, 79], we make each digit class an anomaly and consider the remaining 9 digits as normal examples. Our model is trained with only normal data and tested with both normal and anomalous data. We compare with the BiGAN-based anomaly detection [79], MEG [37] and VAE using area under the precision-recall curve (AUPRC) as in [79]. Table 4 shows the results.

| Heldout Digit | 1 | 4 | 5 | 7 | 9 |
|---|---|---|---|---|---|
| VAE | 0.063 | 0.337 | 0.325 | 0.148 | 0.104 |
| MEG | $0.281 \pm 0.035$ | $0.401 \pm 0.061$ | $0.402 \pm 0.062$ | $0.290 \pm 0.040$ | $0.342 \pm 0.034$ |
| BiGAN-$\sigma$ | $0.287 \pm 0.023$ | $0.443 \pm 0.029$ | $0.514 \pm 0.029$ | $0.347 \pm 0.017$ | $0.307 \pm 0.028$ |
| Ours | $\mathbf{0.336 \pm 0.008}$ | $\mathbf{0.630 \pm 0.017}$ | $\mathbf{0.619 \pm 0.013}$ | $\mathbf{0.463 \pm 0.009}$ | $\mathbf{0.413 \pm 0.010}$ |

Table 4: AUPRC scores for unsupervised anomaly detection on MNIST. Numbers are taken from [37] and results for our model are averaged over last 10 epochs to account for variance.

## 3.5 Computational cost

Our method involving MCMC sampling is more costly than VAEs with amortized inference. Our model is approximately 4 times slower than VAEs on image datasets. On text datasets, ours does not have an disadvantage compared to VAEs on total training time (despite longer per-iteration time) because of better posterior samples from short run MCMC than amortized inference and the overhead of the techniques that VAEs take to address posterior collapse. To test our method's scalability, we trained a larger generator on CelebA ($128 \times 128$). It produced faithful samples (see Figure 1).

# 4 Discussion and conclusion

## 4.1 Modeling strategies and related work

We now put our work within the bigger picture of modeling and learning, and discuss related work.

**Energy-based model and top-down generation model**. A top-down model or a directed acyclic graphical model is of a simple factorized form that is capable of ancestral sampling. The prototype of such a model is factor analysis [61], which has been generalized to independent component analysis [32], sparse coding [53], non-negative matrix factorization [39], etc. An early example of a multi-layer top-down model is the generation model of Helmholtz machine [29]. An EBM defines an unnormalized density or a Gibbs distribution. The prototypes of such a model are exponential family distribution, the Boltzmann machine [1, 30, 62, 40], and the FRAME (Filters, Random field, And Maximum Entropy) model [83]. [81] contrasted these two classes of models, calling the top-down latent variable model the generative model, and the energy-based model the descriptive model. [23] proposed to integrate the two models, where the top-down generation model generates textons, while the EBM prior accounts for the perceptual organization or Gestalt laws of textons. Our model follows

such a plan. Recently, DVAEs [60, 71, 70] adopted restricted Boltzmann machines as the prior model for binary latent variables and a deep neural network as the top-down generation model.

The energy-based model can be translated into a classifier and vice versa via the Bayes rule [24, 68, 12, 75, 33, 38, 19, 22, 55]. The energy function in the EBM can be viewed as an objective function, a cost function, or a critic [63]. It captures regularities, rules or constrains. It is easy to specify, although optimizing or sampling the energy function requires iterative computation such as MCMC. The maximum likelihood learning of EBM can be interpreted as an adversarial scheme [77, 76, 73, 28, 17], where the MCMC serves as a generator or an actor and the energy function serves as an evaluator or a critic. The top-down generation model can be viewed as an actor [63] that directly generates samples. It is easy to sample from, though a complex top-down model is necessary for high quality samples. Comparing the two models, the scalar-valued energy function can be more expressive than the vector-valued top-down network of the same complexity, while the latter is much easier to sample from. It is thus desirable to let EBM take over the top layers of the top-down model to make it more expressive and make EBM learning feasible.

**Energy-based correction of top-down model**. The top-down model usually assumes independent nodes at the top layer and conditional independent nodes at subsequent layers. We can introduce energy terms at multiple layers to correct the independence or conditional independence assumptions, and to introduce inductive biases. This leads to a latent energy-based model. However, unlike undirected latent EBM, the energy-based correction is learned on top of a directed top-down model, and this can be easier than learning an undirected latent EBM from scratch. Our work is a simple example of this strategy where we correct the prior distribution. We can also correct the generation model in the data space.

**From data space EBM to latent space EBM**. EBM learned in data space such as image space [49, 43, 75, 18, 26, 51, 16] can be highly multi-modal, and MCMC sampling can be difficult. We can introduce latent variables and learn an EBM in latent space, while also learning a mapping from the latent space to the data space. Our work follows such a strategy. Earlier papers on this strategy are [81, 23, 4, 7, 37]. Learning EBM in latent space can be much more feasible than in data space in terms of MCMC sampling, and much of past work on EBM can be recast in the latent space.

**Short-run MCMC and amortized computation**. Recently, [51] proposed to use short-run MCMC to sample from the EBM in data space. [52] used it to sample the latent variables of a top-down generation model from their posterior distribution. [31] used it to improve the posterior samples from an inference network. Our work adopts short-run MCMC to sample from both the prior and the posterior of the latent variables. We provide theoretical foundation for the learning algorithm with short-run MCMC sampling. Our theoretical formulation can also be used to jointly train networks that amortize the MCMC sampling from the posterior and prior distributions.

**Generator model with flexible prior**. The expressive power of the generator network for image and text generation comes from the top-down network that maps a simple prior to be close to the data distribution. Most of the existing papers [45, 65, 2, 13, 69, 37] assume that the latent vector follows a given simple prior, such as isotropic Gaussian distribution or uniform distribution. However, such assumption may cause ineffective generator learning as observed in [11, 67]. Some VAE variants attempted to address the mismatch between the prior and the aggregate posterior. VampPrior [66] parameterized the prior based on the posterior inference model, while [3] proposed to construct priors using rejection sampling. ARAE [80] learned an implicit prior with adversarial training. Recently, some papers used two-stage approach [10, 20]. They first trained a VAE or deterministic auto-encoder. To enable generation from the model, they fitted a VAE or Gaussian mixture to the posterior samples from the first-stage model. VQ-VAE [72] adopted a similar approach and an autoregressive distribution over $z$ was learned from the posterior samples. All of these prior models generally follow the empirical Bayes philosophy, which is also one motivation of our work.

## 4.2  Conclusion

EBM has many applications, however, its soundness and its power are limited by the difficulty with MCMC sampling. By moving from data space to latent space, and letting the EBM stand on an expressive top-down network, MCMC-based learning of EBM becomes sound and feasible, and EBM in latent space can capture regularities in data effectively. We may unleash the power of EBM in the latent space for many applications.

## Broader Impact

Our work can be of interest to researchers working on generator model, energy-based models, MCMC sampling and unsupervised learning. It may also be of interest to people who are interested in image synthesis and text generation.

## Acknowledgments and Disclosure of Funding

We thank the four reviewers for their insightful comments and useful suggestions. The work is supported by NSF DMS-2015577; DARPA XAI project N66001-17-2-4029; ARO project W911NF1810296; ONR MURI project N00014-16-1-2007; and XSEDE grant ASC170063. We thank the NVIDIA cooperation for the donation of 2 Titan V GPUs.

## Footnotes

[2]`https://bpucla.github.io/latent-space-ebm-prior-project/`

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
