[Supplementary Material]

# Supplementary Materials for Learning Latent Space Energy-Based Prior Model

## A   Theoretical derivations

In this section, we shall derive most of the equations in the main text. We take a step by step approach, starting from simple identities or results, and gradually reaching the main results. Our derivations are unconventional, but they pertain more to our model and learning method.

### A.1   A simple identity

Let $x \sim p_\theta(x)$. A useful identity is

$$E_\theta[\nabla_\theta \log p_\theta(x)] = 0, \tag{1}$$

where $E_\theta$ (or $E_{p_\theta}$) is the expectation with respect to $p_\theta$.

The proof is one liner:

$$E_\theta[\nabla_\theta \log p_\theta(x)] = \int [\nabla_\theta \log p_\theta(x)]p_\theta(x)dx = \int \nabla_\theta p_\theta(x)dx = \nabla_\theta \int p_\theta(x)dx = \nabla_\theta 1 = 0. \tag{2}$$

The above identity has generalized versions, such as the one underlying the policy gradient [18, 17], $\nabla_\theta E_\theta[R(x)] = E_\theta[R(x)\nabla_\theta \log p_\theta(x)]$. By letting $R(x) = 1$, we get (1).

### A.2   Maximum likelihood estimating equation

The simple identity (1) also underlies the consistency of MLE. Suppose we observe $(x_i, i = 1, ..., n) \sim p_{\theta_{\text{true}}}(x)$ independently, where $\theta_{\text{true}}$ is the true value of $\theta$. The log-likelihood is

$$L(\theta) = \frac{1}{n} \sum_{i=1}^{n} \log p_\theta(x_i). \tag{3}$$

The maximum likelihood estimating equation is

$$L'(\theta) = \frac{1}{n} \sum_{i=1}^{n} \nabla_\theta \log p_\theta(x_i) = 0. \tag{4}$$

According to the law of large number, as $n \to \infty$, the above estimating equation converges to

$$E_{\theta_{\text{true}}}[\nabla_\theta \log p_\theta(x)] = 0, \tag{5}$$

where $\theta$ is the unknown value to be solved, while $\theta_{\text{true}}$ is fixed. According to the simple identity (1), $\theta = \theta_{\text{true}}$ is the solution to the above estimating equation (5), no matter what $\theta_{\text{true}}$ is. Thus with regularity conditions, such as identifiability of the model, the MLE converges to $\theta_{\text{true}}$ in probability.

The optimality of the maximum likelihood estimating equation among all the asymptotically unbiased estimating equations can be established based on a further generalization of the simple identity (1).

We shall justify our learning method with short-run MCMC in terms of an estimating equation, which is a perturbation of the maximum likelihood estimating equation (4).

## A.3 MLE learning gradient for $\theta$

Recall that $p_\theta(x, z) = p_\alpha(z)p_\beta(x|z)$, where $\theta = \{\alpha, \beta\}$. The learning gradient for an observation $x$ is as follows:

$$\nabla_\theta \log p_\theta(x) = \mathrm{E}_{p_\theta(z|x)}\left[\nabla_\theta \log p_\theta(x, z)\right] = \mathrm{E}_{p_\theta(z|x)}\left[\nabla_\theta(\log p_\alpha(z) + \log p_\beta(x|z))\right]. \quad (6)$$

The above identity is a simple consequence of the simple identity (1).

$$\mathrm{E}_{p_\theta(z|x)}\left[\nabla_\theta \log p_\theta(x, z)\right] = \mathrm{E}_{p_\theta(z|x)}\left[\nabla_\theta \log p_\theta(z|x) + \nabla_\theta \log p_\theta(x)\right] \quad (7)$$

$$= \mathrm{E}_{p_\theta(z|x)}\left[\nabla_\theta \log p_\theta(z|x)\right] + \mathrm{E}_{p_\theta(z|x)}\left[\nabla_\theta \log p_\theta(x)\right] \quad (8)$$

$$= 0 + \nabla_\theta \log p_\theta(x), \quad (9)$$

because of the fact that $\mathrm{E}_{p_\theta(z|x)}\left[\nabla_\theta \log p_\theta(z|x)\right] = 0$ according to the simple identity (1), while $\mathrm{E}_{p_\theta(z|x)}\left[\nabla_\theta \log p_\theta(x)\right] = \nabla_\theta \log p_\theta(x)$ because what is inside the expectation only depends on $x$, but does not depend on $z$.

The above identity (6) is related to the EM algorithm [2], where $x$ is the observed data, $z$ is the missing data, and $\log p_\theta(x, z)$ is the complete-data log-likelihood.

## A.4 MLE learning gradient for $\alpha$

For the prior model $p_\alpha(z) = \frac{1}{Z(\alpha)}\exp(f_\alpha(z))p_0(z)$, we have $\log p_\alpha(z) = f_\alpha(z) - \log Z(\alpha) + \log p_0(z)$. Applying the simple identity (1), we have

$$\mathrm{E}_\alpha[\nabla_\alpha \log p_\alpha(z)] = \mathrm{E}_\alpha[\nabla_\alpha f_\alpha(z) - \nabla_\alpha \log Z(\alpha)] = \mathrm{E}_\alpha[\nabla_\alpha f_\alpha(z)] - \nabla_\alpha \log Z(\alpha) = 0. \quad (10)$$

Thus

$$\nabla_\alpha \log Z(\alpha) = \mathrm{E}_\alpha[\nabla_\alpha f_\alpha(z)]. \quad (11)$$

Hence the derivative of the log-likelihood is

$$\nabla_\alpha \log p_\alpha(x) = \nabla_\alpha f_\alpha(z) - \nabla_\alpha \log Z(\alpha) = \nabla_\alpha f_\alpha(z) - \mathrm{E}_\alpha[\nabla_\alpha f_\alpha(z)]. \quad (12)$$

According to equation (6) in the previous subsection, the learning gradient for $\alpha$ is

$$\nabla_\alpha \log p_\theta(x) = \mathrm{E}_{p_\theta(z|x)}\left[\nabla_\alpha \log p_\alpha(z)\right] \quad (13)$$

$$= \mathrm{E}_{p_\theta(z|x)}[\nabla_\alpha f_\alpha(z) - \mathrm{E}_{p_\alpha(z)}[\nabla_\alpha f_\alpha(z))]] \quad (14)$$

$$= \mathrm{E}_{p_\theta(z|x)}[\nabla_\alpha f_\alpha(z)] - \mathrm{E}_{p_\alpha(z)}[\nabla_\alpha f_\alpha(z)]. \quad (15)$$

## A.5 Re-deriving simple identity in terms of $D_{KL}$

We shall provide a theoretical understanding of the learning method with short-run MCMC in terms of Kullback-Leibler divergences. We start from some simple results.

The simple identity (1) also follows from Kullback-Leibler divergence. Consider

$$D(\theta) = D_{KL}(p_{\theta_*}(x)\|p_\theta(x)), \quad (16)$$

as a function of $\theta$ with $\theta_*$ fixed. Suppose the model $p_\theta$ is identifiable, then $D(\theta)$ achieves its minimum 0 at $\theta = \theta_*$, thus $D'(\theta_*) = 0$. Meanwhile,

$$D'(\theta) = -\mathrm{E}_{\theta_*}[\nabla_\theta \log p_\theta(x)]. \quad (17)$$

Thus

$$\mathrm{E}_{\theta_*}[\nabla_\theta \log p_{\theta_*}(x)] = 0. \quad (18)$$

Since $\theta_*$ is arbitrary in the above derivation, we can replace it by a generic $\theta$, i.e.,

$$\mathrm{E}_\theta[\nabla_\theta \log p_\theta(x)] = 0, \quad (19)$$

which is the simple identity (1).

As a notational convention, for a function $f(\theta)$, we write $f'(\theta_*) = \nabla_\theta f(\theta_*)$, i.e., the derivative of $f(\theta)$ at $\theta_*$.

## A.6 Re-deriving MLE learning gradient in terms of perturbation by $D_{KL}$ terms

We now re-derive MLE learning gradient in terms of perturbation of log-likelihood by Kullback-Leibler divergence terms. Then the learning method with short-run MCMC can be easily understood.

At iteration $t$, fixing $\theta_t$, we want to calculate the gradient of the log-likelihood function for an observation $x$, $\log p_\theta(x)$, at $\theta = \theta_t$. Consider the following computationally tractable perturbation of the log-likelihood

$$l_\theta(x) = \log p_\theta(x) - D_{KL}(p_{\theta_t}(z|x)\|p_\theta(z|x)) + D_{KL}(p_{\alpha_t}(z)\|p_\alpha(z)). \tag{20}$$

In the above, as a function of $\theta$, with $\theta_t$ fixed, $D_{KL}(p_{\theta_t}(z|x)\|p_\theta(z|x))$ is minimized at $\theta = \theta_t$, thus its derivative at $\theta_t$ is 0. As a function of $\alpha$, with $\alpha_t$ fixed, $D_{KL}(p_{\alpha_t}(z)\|p_\alpha(z))$ is minimized at $\alpha = \alpha_t$, thus its derivative at $\alpha_t$ is 0. Thus

$$\nabla_\theta \log p_{\theta_t}(x) = \nabla_\theta l_{\theta_t}(x). \tag{21}$$

We now unpack $l_\theta(x)$ to see that it is computationally tractable, and we can obtain its derivative at $\theta_t$.

$$\nabla_\theta l_\theta(x) = \log p_\theta(x) + E_{p_{\theta_t}(z|x)}[\log p_\theta(z|x)] - E_{p_{\alpha_t}(z)}[\log p_\alpha(z)] + c \tag{22}$$

$$= E_{p_{\theta_t}(z|x)}[\log p_\theta(x,z)] - E_{p_{\alpha_t}(z)}[\log p_\alpha(z)] + c \tag{23}$$

$$= E_{p_{\theta_t}(z|x)}[\log p_\alpha(z) + \log p_\beta(x|z)] - E_{p_{\alpha_t}(z)}[\log p_\alpha(z)] + c \tag{24}$$

$$= E_{p_{\theta_t}(z|x)}[\log p_\alpha(z)] - E_{p_{\alpha_t}(z)}[\log p_\alpha(z)] + E_{p_{\theta_t}(z|x)}[\log p_\beta(x|z)] + c \tag{25}$$

$$= E_{p_{\theta_t}(z|x)}[f_\alpha(z)] - E_{p_{\alpha_t}(z)}[f_\alpha(z)] + E_{p_{\theta_t}(z|x)}[\log p_\beta(x|z)] + c + c', \tag{26}$$

where $\log Z(\alpha)$ term gets canceled,

$$c = -E_{p_{\theta_t}(z|x)}[\log p_{\theta_t}(z|x)] + E_{p_{\alpha_t}(z)}[\log p_{\alpha_t}(z)], \tag{27}$$

$$c' = E_{p_{\theta_t}(z|x)}[\log p_0(z)] - E_{p_{\alpha_t}(z)}[\log p_0(z)], \tag{28}$$

do not depend on $\theta$. $c$ consists of two entropy terms. Now taking derivative at $\theta_t$, we have

$$\delta_{\alpha_t}(x) = \nabla_\alpha l(\theta_t) = E_{p_{\theta_t}(z|x)}[\nabla_\alpha f_{\alpha_t}(z)] - E_{p_{\alpha_t}(z)}[\nabla_\alpha f_{\alpha_t}(z)], \tag{29}$$

$$\delta_{\beta_t}(x) = \nabla_\beta l(\theta_t) = E_{p_{\theta_t}(z|x)}[\nabla_\beta \log p_{\beta_t}(x|z)]. \tag{30}$$

Averaging over the observed examples $\{x_i, i = 1, ..., n\}$ leads to MLE learning gradient.

In the above, we calculate the gradient of $\log p_\theta(x)$ at $\theta_t$. Since $\theta_t$ is arbitrary in the above derivation, if we replace $\theta_t$ by a generic $\theta$, we get the gradient of $\log p_\theta(x)$ at a generic $\theta$, i.e.,

$$\delta_\alpha(x) = \nabla_\alpha \log p_\theta(x) = E_{p_\theta(z|x)}[\nabla_\alpha f_\alpha(z)] - E_{p_\alpha(z)}[\nabla_\alpha f_\alpha(z)], \tag{31}$$

$$\delta_\beta(x) = \nabla_\beta \log p_\theta(x) = E_{p_\theta(z|x)}[\nabla_\beta \log p_\beta(x|z)]. \tag{32}$$

The above calculations are related to the EM algorithm [2] and the learning of energy-based model.

In EM algorithm, the complete-data log-likelihood $Q$ serves as a surrogate for the observed-data log-likelihood $\log p_\theta(x)$, where

$$Q(\theta|\theta_t) = \log p_\theta(x) - D_{KL}(p_{\theta_t}(z|x)\|p_\theta(z|x)), \tag{33}$$

and $\theta_{t+1} = \arg\max_\theta Q(\theta|\theta_t)$, where $Q(\theta|\theta_t)$ is a lower-bound of $\log p_\theta(x)$ or minorizes the latter. $Q(\theta|\theta_t)$ and $\log p_\theta(x)$ touch each other at $\theta_t$, and they are co-tangent at $\theta_t$. Thus the derivative of $\log p_\theta(x)$ at $\theta_t$ is the same as the derivative of $Q(\theta|\theta_t)$ at $\theta = \theta_t$.

In EBM, $D_{KL}(p_{\alpha_t}(z)\|p_\alpha(z))$ serves to cancel $\log Z(\alpha)$ term in the EBM prior, and is related to the second divergence term in contrastive divergence [6].

## A.7 Maximum likelihood estimating equation for $\theta = (\alpha, \beta)$

The MLE estimating equation is

$$\frac{1}{n}\sum_{i=1}^n \nabla_\theta \log p_\theta(x_i) = 0. \tag{34}$$

Based on (31) and (32), the estimating equation is

$$\frac{1}{n}\sum_{i=1}^{n}\delta_{\alpha}(x_i) = \frac{1}{n}\sum_{i=1}^{n}\mathrm{E}_{p_\theta(z_i|x_i)}[\nabla_\alpha f_\alpha(z_i)] - \mathrm{E}_{p_\alpha(z)}[\nabla_\alpha f_\alpha(z)] = 0, \tag{35}$$

$$\frac{1}{n}\sum_{i=1}^{n}\delta_{\beta}(x_i) = \frac{1}{n}\sum_{i=1}^{n}\mathrm{E}_{p_\theta(z_i|x_i)}[\nabla_\beta \log p_\beta(x_i|z_i)] = 0. \tag{36}$$

## A.8 Learning with short-run MCMC as perturbation of log-likelihood

Based on the above derivations, we can see that learning with short-run MCMC is also a perturbation of log-likelihood, except that we replace $p_{\theta_t}(z|x)$ by $\tilde{p}_{\theta_t}(z|x)$, and replace $p_{\alpha_t}(z)$ by $\tilde{p}_{\alpha_t}(z)$, where $\tilde{p}_{\theta_t}(z|x)$ and $\tilde{p}_{\alpha_t}(z)$ are produced by short-run MCMC.

At iteration $t$, fixing $\theta_t$, the updating rule based on short-run MCMC follows the gradient of the following function, which is a perturbation of log-likelihood for the observation $x$,

$$\tilde{l}_\theta(x) = \log p_\theta(x) - D_{KL}(\tilde{p}_{\theta_t}(z|x)\|p_\theta(z|x)) + D_{KL}(\tilde{p}_{\alpha_t}(z)\|p_\alpha(z)). \tag{37}$$

The above is a function of $\theta$, while $\theta_t$ is fixed.

In full parallel to the above subsection, we have

$$\tilde{l}_\theta(x) = \mathrm{E}_{\tilde{p}_{\theta_t}(z|x)}[f_\alpha(z)] - \mathrm{E}_{\tilde{p}_{\alpha_t}(z)}[f_\alpha(z)] + \mathrm{E}_{\tilde{p}_{\theta_t}(z|x)}[\log p_\beta(x|z)] + c + c', \tag{38}$$

where $c$ and $c'$ do not depend on $\theta$. Thus, taking derivative of the function $\tilde{l}_\theta(x)$ at $\theta = \theta_t$, we have

$$\tilde{\delta}_{\alpha_t}(x) = \nabla_\alpha \tilde{l}(\theta_t) = \mathrm{E}_{\tilde{p}_{\theta_t}(z|x)}[\nabla_\alpha f_{\alpha_t}(z)] - \mathrm{E}_{\tilde{p}_{\alpha_t}(z)}[\nabla_\alpha f_{\alpha_t}(z)], \tag{39}$$

$$\tilde{\delta}_{\beta_t}(x) = \nabla_\beta \tilde{l}(\theta_t) = \mathrm{E}_{\tilde{p}_{\theta_t}(z|x)}[\nabla_\beta \log p_{\beta_t}(x|z)]. \tag{40}$$

Averaging over $\{x_i, i = 1, ..., n\}$, we get the updating rule based on short-run MCMC. That is, the learning rule based on short-run MCMC follows the gradient of a perturbation of the log-likelihood function where the perturbations consists of two $D_{KL}$ terms.

$D_{KL}(\tilde{p}_{\theta_t}(z|x)\|p_\theta(z|x))$ is related to VAE [10], where $\tilde{p}_{\theta_t}(z|x)$ serves as an inference model, except that we do not learn a separate inference network. $D_{KL}(\tilde{p}_{\alpha_t}(z)\|p_\alpha(z))$ is related to contrastive divergence [6], except that $\tilde{p}_{\alpha_t}(z)$ is initialized from the Gaussian white noise $p_0(z)$, instead of the data distribution of observed examples.

$D_{KL}(\tilde{p}_{\theta_t}(z|x)\|p_\theta(z|x))$ and $D_{KL}(\tilde{p}_{\alpha_t}(z)\|p_\alpha(z))$ cause the bias relative to MLE learning. MLE is impractical because we cannot do exact sampling with MCMC.

However, the bias may not be all that bad. In learning $\beta$, $D_{KL}(\tilde{p}_{\theta_t}(z|x)\|p_\theta(z|x))$ may force the model to be biased towards the approximate short-run posterior $\tilde{p}_{\theta_t}(z|x)$, so that the short-run posterior is close to the true posterior. In learning $\alpha$, the update based on $\mathrm{E}_{\tilde{p}_\theta(z|x)}[\nabla_\alpha f_\alpha(z)] - \mathrm{E}_{\tilde{p}_\alpha(z)}[\nabla_\alpha f_\alpha(z)]$ may force the short-run prior $\tilde{p}_\alpha(z)$ to match the short-run posterior $\tilde{p}_\theta(z|x)$.

## A.9 Perturbation of maximum likelihood estimating equation

The fixed point of the learning algorithm based on short-run MCMC is where the update is 0, i.e.,

$$\frac{1}{n}\sum_{i=1}^{n}\tilde{\delta}_{\alpha}(x_i) = \frac{1}{n}\sum_{i=1}^{n}\mathrm{E}_{\tilde{p}_\theta(z_i|x_i)}[\nabla_\alpha f_\alpha(z_i)] - \mathrm{E}_{\tilde{p}_\alpha(z)}[\nabla_\alpha f_\alpha(z)] = 0, \tag{41}$$

$$\frac{1}{n}\sum_{i=1}^{n}\tilde{\delta}_{\beta}(x_i) = \frac{1}{n}\sum_{i=1}^{n}\mathrm{E}_{\tilde{p}_\theta(z_i|x_i)}[\nabla_\beta \log p_\beta(x_i|z_i)] = 0. \tag{42}$$

This is clearly a perturbation of the MLE estimating equation in (35) and (36). The above estimating equation defines an estimator, where the learning algorithm with short-run MCMC converges.

## A.10 Three $D_{KL}$ terms

We can rewrite the objective function (37) in a more revealing form. Let $(x_i, i = 1, ..., n) \sim p_{\text{data}}(x)$ independently, where $p_{\text{data}}(x)$ is the data distribution. At time step $t$, with fixed $\theta_t$, learning based on short-run MCMC follows the gradient of

$$\frac{1}{n}\sum_{i=1}^{n}[\log p_\theta(x_i) - D_{KL}(\tilde{p}_{\theta_t}(z_i|x_i)\|p_\theta(z_i|x_i)) + D_{KL}(\tilde{p}_{\alpha_t}(z)\|p_\alpha(z))]. \tag{43}$$

Let us assume $n$ is large enough, so that the average is practically the expectation with respect to $p_{\text{data}}$. Then MLE maximizes $\frac{1}{n}\sum_{i=1}^{n}\log p_\theta(x_i) \doteq \mathrm{E}_{p_{\text{data}}(x)}[\log p_\theta(x)]$, which is equivalent to minimizing $D_{KL}(p_{\text{data}}(x)\|p_\theta(x))$. The learning with short-run MCMC follows the gradient that minimizes

$$D_{KL}(p_{\text{data}}(x)\|p_\theta(x)) + D_{KL}(\tilde{p}_{\theta_t}(z|x)\|p_\theta(z|x)) - D_{KL}(\tilde{p}_{\alpha_t}(z)\|p_\alpha(z)), \tag{44}$$

where, with some abuse of notation, we now define

$$D_{KL}(\tilde{p}_{\theta_t}(z|x)\|p_\theta(z|x)) = \mathrm{E}_{p_{\text{data}}(x)}\mathrm{E}_{\tilde{p}_{\theta_t}(z|x)}\left[\log\frac{\tilde{p}_{\theta_t}(z|x)}{p_\theta(z|x)}\right], \tag{45}$$

where we also average over $x \sim p_{\text{data}}(x)$, instead fixing $x$ as before.

The objective (44) is clearly a perturbation of the MLE, as the MLE is based on the first $D_{KL}$ in (44). The signs in front of the remaining two $D_{KL}$ perturbations also become clear. The sign in front of $D_{KL}(\tilde{p}_{\theta_t}(z|x)\|p_\theta(z|x))$ is positive because

$$D_{KL}(p_{\text{data}}(x)\|p_\theta(x)) + D_{KL}(\tilde{p}_{\theta_t}(z|x)\|p_\theta(z|x)) = D_{KL}(p_{\text{data}}(x)\tilde{p}_{\theta_t}(z|x)\|p_\alpha(x)p_\beta(x|z)), \tag{46}$$

where the $D_{KL}$ on the right hand side is about the joint distributions of $(x, z)$, and is more tractable than the first $D_{KL}$ on the left hand side, which is for MLE. This underlies EM and VAE. Now subtracting the third $D_{KL}$, we have the following special form of contrastive divergence

$$D_{KL}(p_{\text{data}}(x)\tilde{p}_{\theta_t}(z|x)\|p_\alpha(z)p_\beta(x|z)) - D_{KL}(\tilde{p}_{\alpha_t}(z)\|p_\alpha(z)), \tag{47}$$

where the negative sign in front of $D_{KL}(\tilde{p}_{\alpha_t}(z)\|p_\alpha(z))$ is to cancel the intractable $\log Z(\alpha)$ term.

The above contrastive divergence also has an adversarial interpretation. When $p_\alpha(z)$ or $\alpha$ is updated, $p_\alpha(z)p_\beta(x|z)$ gets closer to $p_{\text{data}}(x)\tilde{p}_{\theta_t}(z|x)$, while getting away from $\tilde{p}_{\alpha_t}(z)$, i.e., $p_\alpha$ seeks to criticize the samples from $\tilde{p}_{\alpha_t}(z)$ by comparing them to the posterior samples of $z$ inferred from the real data.

As mentioned in the main text, we can also exponentially tilt $p_0(x, z) = p_0(z)p_\beta(x|z)$ to $p_\theta(x, z) = \frac{1}{Z(\theta)}\exp(f_\alpha(x, z))p_0(x, z)$, or equivalently, exponentially tilt $p_0(z, \epsilon) = p_0(z)p(\epsilon)$. The above derivations can be easily adapted to such a model, which we choose not to explore due to the complexity of EBM in the data space.

## A.11 Amortized inference and synthesis networks

We can jointly train two extra networks together with the original model to amortize the short-run MCMC for inference and synthesis sampling. Specifically, we use an inference network $q_\phi(z|x)$ to amortize the short-run MCMC that produces $\tilde{p}_\theta(z|x)$, and we use a synthesis network $q_\psi(z)$ to amortize the short-run MCMC that produces $\tilde{p}_\alpha(z)$.

We can then define the following objective function in parallel with the objective function (44) in the above subsection,

$$\Delta(\theta, \phi, \psi) = D_{KL}(p_{\text{data}}(x)\|p_\theta(x)) + D_{KL}(q_\phi(z|x)\|p_\theta(z|x)) - D_{KL}(q_\psi(z)\|p_\alpha(z)), \tag{48}$$

and we can jointly learn $\theta$, $\phi$ and $\psi$ by

$$\min_\theta \min_\phi \max_\psi \Delta(\theta, \phi, \psi). \tag{49}$$

See [4, 5] for related formulations. The learning of the inference network $q_\phi(z|x)$ follows VAE. The learning of the synthesis network $q_\psi(z)$ is based on variational approximation to $p_\alpha(z)$. The pair $p_\alpha(z)$ and $q_\psi(z)$ play adversarial roles, where $q_\psi(z)$ serves as an actor and $p_\alpha(z)$ serves as a critic.

# B  Experiments

## B.1  Experiment details

**Data.** Image datasets include SVHN [15] $(32 \times 32 \times 3)$, CIFAR-10 [11] $(32 \times 32 \times 3)$, and CelebA [13] $(64 \times 64 \times 3)$. We use the full training split of SVHN $(73, 257)$ and CIFAR-10 $(50, 000)$ and take $40, 000$ examples of CelebA as training data following [16]. The training images are resized and scaled to $[-1, 1]$. Text datasets include PTB [14], Yahoo [19], and SNLI [1], following recent work on text generative modeling with latent variables [8, 20, 12].

**Model architectures.** The architecture of the EBM, $f_\alpha(z)$, is displayed in Table 2. For text data, the dimensionality of $z$ is set to 32. The generator architectures for the image data are also shown in Table 2. The generators for the text data are implemented with a one-layer unidirectional LSTM [7] and Table 3 lists the number of word embeddings and hidden units of the generators for each dataset.

**Short run dynamics.** The hyperparameters for the short run dynamics are depicted in Table 1 where $K_0$ and $K_1$ denote the number of prior and posterior sampling steps with step sizes $s_0$ and $s_1$, respectively. These are identical across models and data modalities, except for the model for CIFAR-10 which is using $K_1 = 40$ steps.

| Short Run Dynamics Hyperparameters | |
|---|---|
| Hyperparameter | Value |
| $K_0$ | 60 |
| $s_0$ | 0.4 |
| $K_1$ | 20 |
| $s_1$ | 0.1 |

Table 1: Hyperparameters for short run dynamics.

**Optimization.** The parameters for the EBM and image generators are initialized with Xavier normal [3] and those for the text generators are initialized from a uniform distribution, $\text{Unif}(-0.1, 0.1)$, following [8, 12]. Adam [9] is adopted for all model optimization. The models are trained until convergence (taking approximately $70, 000$ and $40, 000$ parameter updates for image and text models, respectively).

| | SNLI | PTB | Yahoo |
|---|---|---|---|
| Word Embedding Size | 256 | 128 | 512 |
| Hidden Size of Generator | 256 | 512 | 1024 |

Table 3: The sizes of word embeddings and hidden units of the generators for SNLI, PTB, and Yahoo.

# C  Ablation study

We investigate a range of factors that are potentially affecting the model performance with SVHN as an example. The highlighted number in Tables 4, 5, and 6 is the FID score reported in the main text and compared to other baseline models. It is obtained from the model with the architecture and hyperparameters specified in Table 1 and Table 2 which serve as the reference configuration for the ablation study.

**Fixed prior.** We examine the expressivity endowed with the EBM prior by comparing it to models with a fixed isotropic Gaussian prior. The results are displayed in Table 4. The model with an EBM prior clearly outperforms the model with a fixed Gaussian prior and the same generator as the reference model. The fixed Gaussian models exhibit an enhancement in performance as the generator complexity increases. They however still have an inferior performance compared to the model with an EBM prior even when the fixed Gaussian prior model has a generator with four times more parameters than that of the reference model.

| EBM Model | | |
|---|---|---|
| Layers | In-Out Size | Stride |
| Input: $z$ | 100 | |
| Linear, LReLU | 200 | - |
| Linear, LReLU | 200 | - |
| Linear | 1 | - |
| **Generator Model for SVHN**, ngf = 64 | | |
| Input: $x$ | 1x1x100 | |
| 4x4 convT(ngf x 8), LReLU | 4x4x(ngf x 8) | 1 |
| 4x4 convT(ngf x 4), LReLU | 8x8x(ngf x 4) | 2 |
| 4x4 convT(ngf x 2), LReLU | 16x16x(ngf x 2) | 2 |
| 4x4 convT(3), Tanh | 32x32x3 | 2 |
| **Generator Model for CIFAR-10**, ngf = 128 | | |
| Input: $x$ | 1x1x128 | |
| 8x8 convT(ngf x 8), LReLU | 8x8x(ngf x 8) | 1 |
| 4x4 convT(ngf x 4), LReLU | 16x16x(ngf x 4) | 2 |
| 4x4 convT(ngf x 2), LReLU | 32x32x(ngf x 2) | 2 |
| 3x3 convT(3), Tanh | 32x32x3 | 1 |
| **Generator Model for CelebA**, ngf = 128 | | |
| Input: $x$ | 1x1x100 | |
| 4x4 convT(ngf x 8), LReLU | 4x4x(ngf x 8) | 1 |
| 4x4 convT(ngf x 4), LReLU | 8x8x(ngf x 4) | 2 |
| 4x4 convT(ngf x 2), LReLU | 16x16x(ngf x 2) | 2 |
| 4x4 convT(ngf x 1), LReLU | 32x32x(ngf x 1) | 2 |
| 4x4 convT(3), Tanh | 64x64x3 | 2 |

Table 2: EBM model architectures for all image and text datasets and generator model architectures for SVHN ($32 \times 32 \times 3$), CIFAR-10 ($32 \times 32 \times 3$), and CelebA ($64 \times 64 \times 3$). convT($n$) indicates a transposed convolutional operation with $n$ output feature maps. LReLU indicates the Leaky-ReLU activation function. The leak factor for LReLU is $0.2$ in EBM and $0.1$ in Generator.

| Model | FID |
|---|---|
| Latent EBM Prior | **29.44** |
| **Fixed Gaussian** | |
| same generator | 43.39 |
| generator with 2 times as many parameters | 41.10 |
| generator with 4 times as many parameters | 39.50 |

Table 4: Comparison of the models with a latent EBM prior versus a fixed Gaussian prior. The highlighted number is the reported FID for SVHN and compared to other baseline models in the main text.

**MCMC steps.** We also study how the number of short run MCMC steps for prior inference ($K_0$) and posterior inference ($K_1$). The left panel of Table 5 shows the results for $K_0$ and the right panel for $K_1$. As the number of MCMC steps increases, we observe improved quality of synthesis in terms of FID.

| Steps | FID |
|---|---|
| $K_0 = 40$ | 31.49 |
| $K_0 = 60$ | **29.44** |
| $K_0 = 80$ | 28.32 |

| Steps | FID |
|---|---|
| $K_1 = 20$ | **29.44** |
| $K_1 = 40$ | 27.26 |
| $K_1 = 60$ | 26.13 |

Table 5: Influence of the number of prior and posterior short run steps $K_0$ (left) and $K_1$ (right). The highlighted number is the reported FID for SVHN and compared to other baseline models in the main text.

**Prior EBM and generator complexity.** Table 6 displays the FID scores as a function of the number of hidden features of the prior EBM (nef) and the factor of the number of channels of the generator (ngf, also see Table 2). In general, enhanced model complexity leads to improved generation.

| | **nef** | 50 | 100 | 200 |
|---|---|---|---|---|
| | 32 | 32.25 | 31.98 | 30.78 |
| **ngf** | 64 | 30.91 | 30.56 | **29.44** |
| | 128 | 29.12 | 27.24 | 26.95 |

Table 6: Influence of prior and generator complexity. The highlighted number is the reported FID for SVHN and compared to other baseline models in the main text. **nef** indicates the number of hidden features of the prior EBM and **ngf** denotes the factor of the number of channels of the generator (also see Table 2).

## D  PyTorch code

```python
import torch as t, torch.nn as nn
import torchvision as tv, torchvision.transforms as tfm

img_size, batch_size = 32, 100
nz, nc, ndf, ngf = 100, 3, 200, 64
K_0, a_0, K_1, a_1 = 60, 0.4, 40, 0.1
llhd_sigma = 0.3
n_iter = 70000
device = t.device('cuda' if t.cuda.is_available() else 'cpu')

class _G(nn.Module):
    def __init__(self):
        super().__init__()
        self.gen = nn.Sequential(nn.ConvTranspose2d(nz, ngf*8, 4, 1, 0), nn.LeakyReLU(),
            nn.ConvTranspose2d(ngf*8, ngf*4, 4, 2, 1), nn.LeakyReLU(),
            nn.ConvTranspose2d(ngf*4, ngf*2, 4, 2, 1), nn.LeakyReLU(),
            nn.ConvTranspose2d(ngf*2, nc, 4, 2, 1), nn.Tanh())
    def forward(self, z):
        return self.gen(z)

class _E(nn.Module):
    def __init__(self):
        super().__init__()
        self.ebm = nn.Sequential(nn.Linear(nz, ndf), nn.LeakyReLU(0.2),
            nn.Linear(ndf, ndf), nn.LeakyReLU(0.2),
            nn.Linear(ndf, 1))
    def forward(self, z):
        return self.ebm(z.squeeze()).view(-1, 1, 1, 1)

transform = tfm.Compose([tfm.Resize(img_size), tfm.ToTensor(), tfm.Normalize(([0.5]*3), ([0.5]*3)),])
data = t.stack([x[0] for x in tv.datasets.SVHN(root='data/svhn', transform=transform)]).to(device)

G, E = _G().to(device), _E().to(device)
mse = nn.MSELoss(reduction='sum')
optE = t.optim.Adam(E.parameters(), lr=0.00002, betas=(0.5, 0.999))
optG = t.optim.Adam(G.parameters(), lr=0.0001, betas=(0.5, 0.999))

def sample_p_data():
    return data[t.LongTensor(batch_size).random_(0, data.size(0))].detach()

def sample_p_0(n=batch_size):
    return t.randn(*[n, nz, 1, 1]).to(device)

def sample_langevin_prior(z, E):
    z = z.clone().detach().requires_grad_(True)
    for i in range(K_0):
        en = E(z)
        z_grad = t.autograd.grad(en.sum(), z)[0]
        z.data = z.data - 0.5 * a_0 * a_0 * (z_grad + 1.0 / z.data) + a_0 * t.randn_like(z).data
    return z.detach()

def sample_langevin_posterior(z, x, G, E):
    z = z.clone().detach().requires_grad_(True)
    for i in range(K_1):
        x_hat = G(z)
        g_log_lkhd = 1.0 / (2.0 * llhd_sigma * llhd_sigma) * mse(x_hat, x)
        grad_g = t.autograd.grad(g_log_lkhd, z)[0]
        en = E(z)
        grad_e = t.autograd.grad(en.sum(), z)[0]
        z.data = z.data - 0.5 * a_1 * a_1 * (grad_g + grad_e + 1.0 / z.data) + a_1 * t.randn_like(z).data
    return z.detach()

for i in range(n_iter):
    x = sample_p_data()
    z_e_0, z_g_0 = sample_p_0(), sample_p_0()
    z_e_k, z_g_k = sample_langevin_prior(z_e_0, E), sample_langevin_posterior(z_g_0, x, G, E)

    optG.zero_grad()
    x_hat = G(z_g_k.detach())
    loss_g = mse(x_hat, x) / batch_size
    loss_g.backward()
    optG.step()

    optE.zero_grad()
    en_pos, en_neg = E(z_g_k.detach()).mean(), E(z_e_k.detach()).mean()
    loss_e = en_pos - en_neg
    loss_e.backward()
    optE.step()
```