[Reviews · NeurIPS 2020]

Review 1

Summary and Contributions: Update: I have read the rebuttal and am glad to see that the authors are willing to update the paper to include more discussions about prior work. --- The paper discusses a variational EM approach to learning latent variable models, where the prior distribution is a learnable energy-based model and the posterior inference model is obtained by finite MCMC sampling over the true posterior (available via the energy of the joint). The paper demonstrates superior performance compared to existing VAE approaches that aims to use learnable prior in reconstruction and generation.

Strengths: ### Technical Novelty The paper proposes to consider a learnable energy model as the latent prior model. While there are various existing work on learning a latent prior model, framing it as an energy based model seems relatively interesting. The paper is conscious about the difference of "short-run" MCMC chains and the actual posterior, and provides a discussion in section 3.5 about it. ### Empirical evaluation Empirically the proposed methods appears to have better performance than existing methods that uses learnable prior and and more explicit inference model (such as two-stage VAE).

Weaknesses: ### Positioning and comparison with existing work It seems that the authors are keen to describe the method as "maximum likelihood", and the short run Markov chain approach as an approximation to the method. However, one could also interpret the short run Markov chain sampler as an approximation to the true posterior p_\theta(z | x), and interpret the entire approach as EM / coordinate ascent between the inference sampling procedure and the generative model. Therefore, I personally feel like the claims in lines 191-194 about the approach being more general than exact MLE is slightly misleading. If we use the EM perspective, there exists prior work that utilizes MCMC to derive a better posterior inference estimate; see Hoffman's 2017 paper "Learning Deep Latent Gaussian Models with Markov Chain Monte Carlo", where an inference model is augmented with MCMC steps over the true posterior. It seems to be a highly relevant work to this paper. ### Evaluation Surprisingly, for the image generation task, there is no test log-likelihood metric (or lower bounds) that is observed on most VAE models (since the paper is mostly motivated from an MLE perspective). The lower bound seems to be relatively difficult to estimate due to the partition function being not easy to compute, but I think it is not impossible with annealed importance sampling?

Correctness: I have not found any strict incorrectness with regards to the claims and method.

Clarity: The paper is well written and detailed, but perhaps could benefit from bolding the variables that are assumed to be vectors. Moreover, a discussion of the computational efficiency as a variable of K in the main paper could help practitioners in understanding the computational trade-offs of this approach.

Relation to Prior Work: As mentioned in the weakness section, the work misses one important related work [Hoffman et al, 2017] on short run MCMC utilized to improve VAE inference modeling. If we ignore the energy based latent variable, the proposed MLE method is a special case to the aforementioned paper with a trivial initial inference distribution.

Reproducibility: Yes

Additional Feedback:


Review 2

Summary and Contributions: The paper introduces a method to train generative models based on latent vectors while learning a flexible latent distribution alongside the generator neural network. This latent distribution is based on an energy-based correction of a standard Gaussian distribution, the energy value being defined as the output of a secondary neural network. The training of both networks is performed concurently, using short-run MCMC Langevin dynamics to approximate the latent prior p(z) and the latent posterior p(z|x). The paper then provides experimental comparison of their model to several other approaches from the literature both in the context of image generation and text synthesis, showing their model is competitive or performs better.

Strengths: The authors provide a detailed theoretical analysis and justification of their training algorithm. The model is compared to several other known methods on various datasets, and an ablation study is provided in appendix. The computing cost induced by the use of MCMC (as opposed to a standard VAE) is discussed and appears sufficiently low for the approach to be worth considering as a way to improve the quality of generative models. The use of short-run MCMC as an affordable yet efficient approximation for sampling the variational posterior, and the empirical validation that it gives good results even for relatively short run-lenghts is an interesting result.

Weaknesses: The main uncertainty I have is regarding the computing cost of this approach. The use of MCMC Langevin dynamics to estimate sampling the variational posterior implies a significant increase in the number of backpropagation passes through the generator neural network (K+1 passes in total for each mini-batch). On larger generator models this cost can become quite significant. In particular, the authors only provide a numerical comparison of the training times in appendix for their SVHN example, which is the smallest of the models in their comparison. The computational cost induced on text-based auto-regressive models is not presented at all, which I find a little unsatisfying.

Correctness: The paper provides a well-justified model, and compare it to the literature using standard evaluation scores.

Clarity: The paper is clearly written.

Relation to Prior Work: The prior work regarding MCMC energy based models and VAEs with flexible priors is clearly discussed. The main contribution of the paper seems to be the use of MCMC Langevin dynamics to build an Energy-Based model in the latent space of a generator model, rather than directly on data space, and a joint procedure to train both the EBM and the generator network simultaneously.

Reproducibility: Yes

Additional Feedback: l.61 has a typo : "soild" instead of "solid". **Post-rebuttal response** I'll thank the authors for the clarifications regarding the computing cost of their approach. I think this paper presents an interesting approach and could be a good addition to NeurIPS, however I agree with the concerns raised by the other reviewers regarding relation to prior work and comparison with relevant baselines. As such, I'm changing my score to 6.


Review 3

Summary and Contributions: Authors propose energy based generative model that augments a top down model(DAG) with a latent space learnt via EBM that potentially corrects standard noise prior. They modify MLE formulation by using short run MCMC for learning their model. In addition to providing theoretical basis, they also showcase qualitative samples and quantitative results on their setup.

Strengths: 1)At the outset, paper clearly describes their method and is well articulated. 2)Also appreciate the theoretical section that delves deep into the learning mechanism while providing required insights. 3)Supplementary material comprehensive and has all the required details

Weaknesses: Results are not compelling enough to showcase what the model is capable of. It could be important to showcase what the corrected prior/posterior is capable of than standard distributions comparisons with up-to-date literature could be useful

Correctness: yes

Clarity: yes

Relation to Prior Work: work includes relevant literature, but could include more recent research on inference models (both GAN and VAE based) other than just BiGAN

Reproducibility: Yes

Additional Feedback: I have one main concern that experiment section is not comprehensive enough I would appreciate if there are additional experiments to include usefulness of learnt latent space such as semi-supervised or one-shot related results. I was wondering if some details of theoretical results can go into additional material and possible include some intuitive connections with other topdown inference based generative models.


Review 4

Summary and Contributions: This paper proposes a generative model in the form of p(z, x) = p(z) p(x|z) where p(z) is an exponential tilting of a simple Gaussian prior (i.e., the prior is an energy-based model), the likelihood function p(x|z) is a decoding distribution similar to the one in VAEs. This paper proposes to use short-run MCMC sampling to sample from both true posterior and EBM prior distributions during training. The experimental results show that the proposed model outperforms small VAE modes and different variants of VAEs in image generation, text generation, and anomaly detection tasks. 

Strengths: * Novelty: Recently energy-based generative models have gained a lot of momentum in the community. The idea of using an EBM in the latent space of generative models, as proposed in this paper, is an interesting and impactful extension to the current efforts. EBMs for the prior distribution has been examined in the past for binary latent variables (in DVAEs, see below). However, this paper goes beyond simple energy functions and it shows how EBMs modeled by a neural network can be applied to the continuous latent variables. * Theoretical grounding: I found the theoretical discussion in Sec 3.5 very helpful for understanding the error induced by approximate sampling. It is very interesting to see that short-run MCMC can be thought of as a variational bound on log-likelihood where the "variational" distribution is the distribution of samples generated by the short-run MCMC samples. * Empirical Evaluation: The method is examined in several image and text datasets. It has been shown that the proposed model outperforms several prior works including VAEs that use simple Gaussian distribution. * Relevance to the NeurIPS community: The work is highly relevant to the community. 

Weaknesses: * Missing comparison against persistent sampling: This paper proposes to use short-run MCMC to sample from both the prior and true posterior. In practice, since we have only one prior distribution, sampling from the prior can be also done using persistent sampling which often improves the performance of EBMs by a large margin. It's not clear why the proposed method uses short-run MCMC that can potentially mix slowly and can introduce sampling error. Moreover, Eq. 13 shows that sampling error turns the objective into an upper bound on the log-likelihood. This can be dangerous as the model may start increasing the gap between the distribution of approximate samples and the EBM prior by making the distribution harder to sample from. * Practical limitations: While we have a single prior distribution, we have true posterior distributions as many as training data points. This means that in every parameter update, we require sampling from the true posterior distribution per data point. The computational complexity of sampling from the true posterior is much more than the prior as the former requires evaluation of both prior and decoder networks. This paper limits the decoder to small models with a few layers, but in practice when decoders are deep, running 20 MCMC steps requires evaluating a very expensive network 20 times in each training iteration. It is not clear why the proposed model abandons amortized inference for approximating the true posterior. A variational distribution can be easily used to infer the latent variables in an amortized fashion (as done in VAEs and DVAEs). * Claims on the stability of the algorithm: In line 48, it is claimed that training EBMs unlike GANs doesn't suffer from instability. However, as observed by Du & Mordatch NeurIPS 2019 training energy-based models can be unstable when the sampling procedure in the negative phase cannot catch up with sharp energy functions. In my experience with EBMs, this problem can be a big barrier to training EBMs.

Correctness: As long as I can see the derivations and the overall hypotheses seem correct. Regarding the evaluation, it seems that all the comparisons are done internally by comparing the proposed model against re-implementation of the prior art by authors (please correct me if I am wrong). It would be nice if the authors could also compare the method against prior work that uses similar network architectures by reporting results from the corresponding papers. For example, 2-Stage VAE [12] have reported better FID scores for their method originally (FID score for 2-stage VAE on CIFAR 10, original 72.9, reported 109.77)

Clarity: Yes, the paper is well written.

Relation to Prior Work: * I found the main missing prior works that are not discussed in this paper are DVAEs (DVAE [1], DVAE++[2], and DVAE#[3]) that use energy-based Boltzmann machines in the prior of VAEs. This work differs from DVAEs in two aspects: i) The proposed model goes beyond binary latent variables and simple RBM energy function (which were used in DVAEs), and it uses neural networks to define an energy function over continuous variables. ii) The proposed work uses short-run MCMC to samples from true posterior which can have computational disadvantages as discussed in the weaknesses. [1]: Discrete variational autoencoders, ICLR 2017. [2]: Dvae++: Discrete variational autoencoders with overlapping transformations, ICML 2018. [3]: DVAE#: Discrete Variational Autoencoders with Relaxed Boltzmann Priors, NeurIPS 2019. * Recently, Ghosh et al. ICLR 2020 showed that regularized autoencoders can generate high-quality images. Although the proposed method outperforms Ghosh et al. on the CIFAR-10 and CelebA datasets, it would be nice to include this when referred to regularized autoencoders. Also, VQ-VAE can be considered as a 2-stage VAE and is missing in the discussion. 

Reproducibility: Yes

Additional Feedback: **** After rebuttal **** After discussing with all the reviewers, I agree that this paper has its own merits and it is a good addition to NeurIPS. However, given the current concerns regarding the discussion of the previous work, I'm going to stick to my original rating. I believe the stability issues and the efficiency of training for big models can be a good subject for future studies. *************************

[Author Response · NeurIPS 2020]

Thank you for the valuable comments. We first address several general concerns, and then reply to each reviewer.

**G1: Computational cost.** We discussed it in Supplementary 2.3 and will do a more thorough analysis in the revision.
On a single 1080 Ti, our model took 6 hours and VAE 1.5 hours, and thus being 4 times slower, a similar factor
observed on other image datasets. For text models with an autoregressive generator, our model does not have an
obvious disadvantage compared to SOTA VAEs in terms of total training time (despite its longer per-iteration time)
because of better posterior samples from short run MCMC than amortized inference and the overhead of the techniques
that VAEs take to address posterior collapse (a common phenomenon for text VAEs). For the Yahoo dataset trained
with the largest autoregressive generator, on a single 1080 Ti, our model took 14 hours, SA-VAE (combining amortized
inference and MCMC) took 48 hours, FB-VAE took 5 hours (autoencoder pretraining) + 7 hours, and ARAE took 15
hours. Our approach trades feasible computational cost for expressive prior and simple and accurate inference.

**G2: Scalability.** To test our method's scalability, we trained a large generator with 49 million parameters on CelebA
($128 \times 128$) on 4 V100 GPUs. It converged in 12 hours and produced faithful samples (see Figure 1 in Supplementary).

**G3: RAE (Ghosh et al. ICLR 2020).** We apologize for the citation typo. RAE in Table 1 refers to the regularized
autoencoder, a SOTA VAE model, proposed by Ghosh et al. 2020.

**R1.** Thank you for the insightful comments. We will cite and discuss Hoffman 2017. We shall also discuss computa-
tional efficiency in the main text as you suggested.
**Q1: Relations to MLE/EM/VAE.** We shall tune down the claim to "our method is a practical modification of MLE."
In Section 3.5 on theoretical understanding, eqn (13), the first $D_{KL}$ is related to VAE and EM (line 181; see also
Supplementary line 133). We will add more explanations as you suggested.
**Q2: Log-likelihood.** Yes, it is possible to estimate the log-likelihood by AIS. We shall include it in the revision.

**R2.** We deeply appreciate your positive feedback and insightful summary of our work.
**Q1: Computational cost.** Please see **G1** and **G2** above, which hopefully address your concern.

**R4.** We appreciate your positive comments on theoretical section and supplementary content. We shall do our best to
further strengthen experiments as you suggested. We wish you could reconsider your rating.
**Q1: Up-to-date literature (GAN and VAE).** (1) We wish to clarify that the VAE models we compared against are
up-to-date models. 2sVAE and RAE are considered the SOTA VAEs for image modeling. FB-VAE is the SOTA
model for latent-variable-based language modeling. (2) We shall compare with GAN models in the revision. GANs
underperform basic language models on text modeling due to the non-differentiability of text generation (Caccia et al.
ICLR 2020). Moreover, GAN by itself is not equipped with an inference mechanism for inferring latent variables.
**Q2: Advantage of learnable EBM prior.** Compared to standard Gaussian, the learnable EBM prior leads to more
accurate modeling of data distribution, illustrated by the learned model producing faithful (image and text) samples
and being able to detect anomaly samples. Also see Table 4 and line 171 in Supplementary for a direct comparison to
models with standard Gaussian prior.
**Q3: Semi-supervised learning.** Thanks for the suggestion! We are working on semi-supervised learning with our
model and obtained promising results, which will be included in the revision.

**R5.** We appreciate your in-depth questions. We will discuss relevant work, DVAEs and VQ-VAE, and correct the
citation typo for RAE (see **G3**). Also please see **G1** and **G2** above for our responses to computational concerns.
**Q1: Persistent chain (PC).** We will include the comparison to PC in the revision. In our experiments, short run chains
(SRC) in latent space mixes quickly (see Figure 2 in the main text for SRC and Figure 2 in Supplementary for long
run chains) so that $D_{KL}(\tilde{p}_\alpha(z)\|p_\alpha(z))$ can be small. This is a big advantage of EBM in latent space as compared to
EBM in data space. Advantages of SRC over PC are: 1) theoretical underpinning of the learning method with SRC is
cleaner; 2) in both training and testing, the same short run MCMC is used.
**Q2: Amortized inference.** We shall explore amortized inference in revision as you suggested. We did not use amor-
tized inference for the following reasons. (1) We need posterior samples of $z$ to learn the EBM prior $p_\alpha(z)$, and
short run MCMC can be more reliable for generating posterior samples than amortized inference. (2) VAE is not
conveniently applicable because $\log p_\theta(x, z)$ involves intractable $\log Z_\alpha$ term. (3) Short run MCMC in latent space is
simple and reasonably computationally efficient.
**Q3: Stability.** In our experience and as observed by Grathwohl et al. ICLR 2020, training EBMs with PC as done by
Du & Mordatch is less stable than using short run MCMC. Du & Mordatch stabilizes training with spectral normal-
ization and $L_2$ regularization which are not needed in our experiments. Since our EBM is defined in low-dimensional
latent space, short run MCMC is stable and scalable.
**Q4: Baseline implementations.** Prior art results are copied from the published papers if available. For instance, the
best RAE FID scores for CIFAR-10 and CelebA are taken from Table 1 in Ghosh et al. 2020. FID scores for 2sVAE
are also taken from Ghosh et al. given the comparability of their generator architecture to ours. We shall include the
numbers reported in the 2sVAE paper in the revision.

[Meta-Review · NeurIPS 2020]

This paper introduces a variational EM approach to latent variable models with an energy-based models (EBM) for the latent distribution. The work is a nice contribution in a line of work that revisits the use of EBMs. The reviewers praised the technical novelty and the technical justification of the training algorithm. This work is relevant to the NeurIPS and likely to be of particular interest this year. Some reviewers noted that the current manuscript does not sufficiently discuss related work and some comparisons are lacking, noting in particular the DVAE line of work. Most reviewers agreed that the DVAE and related work concerns are not severe enough to recommend rejection, so I am recommending acceptance. It is critical that the authors take into account the reviewer comments and make a particular effort to address the related work concerns in the revised paper.